# Development of a Serum-Free Medium To Aid Large-Scale Production of *Mycoplasma*-Based Therapies

Raul Burgos,[a] Eva Garcia-Ramallo,[a] Daniel Shaw,[a] Maria Lluch-Senar,[a,d,e] Luis Serrano[a,b,c]

[a]Centre for Genomic Regulation (CRG), The Barcelona Institute of Science and Technology, Barcelona, Spain
[b]Universitat Pompeu Fabra (UPF), Barcelona, Spain
[c]ICREA, Barcelona, Spain
[d]Pulmobiotics Ltd., Barcelona, Spain
[e]Basic Sciences Department, Faculty of Medicine and Health Sciences, Universitat Internacional de Catalunya, Sant Cugat del Vallès, Spain

**ABSTRACT** To assist in the advancement of the large-scale production of safe *Mycoplasma* vaccines and other *Mycoplasma*-based therapies, we developed a culture medium free of animal serum and other animal components for *Mycoplasma pneumoniae* growth. By establishing a workflow method to systematically test different compounds and concentrations, we provide optimized formulations capable of supporting serial passaging and robust growth reaching 60 to 70% of the biomass obtained in rich medium. Global transcriptomic and proteomic analysis showed minor physiological changes upon cell culture in the animal component-free medium, supporting its suitability for the production of *M. pneumoniae*-based therapies. The major contributors to growth performance were found to be glucose as a carbon source, glycerol, cholesterol, and phospholipids as a source of fatty acids. Bovine serum albumin or cyclodextrin (in the animal component-free medium) were required as lipid carriers to prevent lipid toxicity. Connaught Medical Research Laboratories medium (CMRL) used to simplify medium preparation as a source of amino acids, nucleotide precursors, vitamins, and other cofactors could be substituted by cysteine. In fact, the presence of protein hydrolysates such as yeastolate or peptones was found to be essential and preferred over free amino acids, except for the cysteine. Supplementation of nucleotide precursors and vitamins is not strictly necessary in the presence of yeastolate, suggesting that this animal origin-free hydrolysate serves as an efficient source for these compounds. Finally, we adapted the serum-free medium formulation to support growth of *Mycoplasma hyopneumoniae*, a swine pathogen for which inactivated whole-cell vaccines are available.

**IMPORTANCE** *Mycoplasma* infections have a significant negative impact on both livestock production and human health. Vaccination is often the first option to control disease and alleviate the economic impact that some *Mycoplasma* infections cause on milk production, weight gain, and animal health. The fastidious nutrient requirements of these bacteria, however, challenges the industrial production of attenuated or inactivated whole-cell vaccines, which depends on the use of animal serum and other animal raw materials. Apart from their clinical relevance, some *Mycoplasma* species have become cellular models for systems and synthetic biology, owing to the small size of their genomes and the absence of a cell wall, which offers unique opportunities for the secretion and delivery of biotherapeutics. This study proposes medium formulations free of serum and animal components with the potential of supporting large-scale production upon industrial optimization, thus contributing to the development of safe vaccines and other *Mycoplasma*-based therapies.

**KEYWORDS** *Mycoplasma*, media, metabolism, serum-free medium, vaccines

Address correspondence to Raul Burgos, raul.burgos@crg.eu, or Luis Serrano, luis.serrano@crg.eu.

The authors declare a conflict of interest. The authors declare that patent applications were filed for the M. pneumoniae (WO2021078935) and M. hyopneumoniae (WO2021078938) serum-free media. We would also like to state that Luis Serrano and Maria Lluch-Senar are co-founders of Pulmobiotics Ltd., and Maria Lluch-Senar is currently an employee of this company.

Mycoplasmas comprise a large group of species capable of colonizing a wide range of organisms, including animals, plants, insects, and humans (1). They are characterized by the lack of a cell wall, which makes them naturally resistant to antibiotics targeting the cell wall, and suitable cell systems for secretion and delivery of biotherapeutics (2–5). Owing to the small size of their genomes, mycoplasmas, and especially *Mycoplasma pneumoniae*, have also become important cellular models for systems biology (6–15). One consequence of this genome reduction is that they maintain few metabolic capabilities, relying on the host-cell environment for much of their nutrition. Consequently, growth of *Mycoplasma* species in axenic conditions has been traditionally difficult and dependent on animal serum.

Several species of *Mycoplasma* act as commensal bacteria, living innocuously with their host as part of the natural flora. However, many other species are pathogens, such as *M. pneumoniae*, which causes atypical pneumonia (16). Indeed, diseases associated with mycoplasmas are an important economic burden, in both human and livestock systems (17–21). In this regard, vaccination has been proven as an efficient strategy to alleviate the economic impact that some *Mycoplasma* infections cause on milk production, weight gain, and animal health (22–24). Some of the most effective vaccines available against *Mycoplasma* infections are live attenuated or inactivated vaccines, in which growth and production of the bacterial strain is required. The culture medium used in the production of these vaccines usually contains complex animal components such as brain heart infusion broth (BHI), beef heart infusion and peptones (PPLO), pancreatic digest of casein, beef extract broth, or animal serum. The use of animal serum in culture medium entails several disadvantages, including low batch-to-batch reproducibility, undefined composition, possible interference with downstream processing, and a considerably increased cost of the medium. In addition, the use of both animal serum and raw materials derived from animals has important safety concerns, since these may contain viruses, antibiotics, endotoxins, and other bioactive molecules. Therefore, the availability of a culture medium that (i) is safe and free of animal components, (ii) supports robust growth, and (iii) allows serial passaging is fundamental in *Mycoplasma* vaccine development or other *Mycoplasma*-based therapies (2–5).

Apart from industry-oriented research, the availability of a defined serum-free medium would also benefit metabolic studies, as the concentration of any component can be adjusted or simply be removed as desired depending on the experimental design. In this regard, a defined medium for *M. pneumoniae* was previously established based on the reconstructed metabolic network of this bacterium (9). This medium allowed cell survival and maintenance but did not support the robust growth and serial passaging required in industry production or certain experimental designs. Although the use of genome-scale metabolic models may be useful to predict the requirement of certain metabolites, they are limited unless there is a complete understanding of the nutritional constraints represented in the cell and environment (25–27). Consequently, inferring optimal levels of the different nutrients, and accounting for possible effects derived from the interactions between them, is still a challenging task.

The aim of this study was to develop a serum-free medium capable of supporting robust growth of *M. pneumoniae* to aid the large-scale production of *Mycoplasma*-based therapies. To this end, we established a workflow method to systematically test the performance of different medium formulations. As a result, we obtained distinct medium compositions free of animal serum (vB10) or totally free of animal-derived components (vB13), capable of supporting growth of up to 60 to 70% of the biomass obtained with rich medium. The new formulations allowed serial passaging and maintained a cell physiology state without major proteomic or transcriptional changes. We also performed single-omission experiments to define strictly essential components required for growth, suggesting a minimal medium formulation that could also assist certain metabolic studies. Finally, we evaluated the performance of the developed medium in *Mycoplasma hyopneumoniae*, which is the primary pathogen of enzootic pneumonia, a chronic respiratory

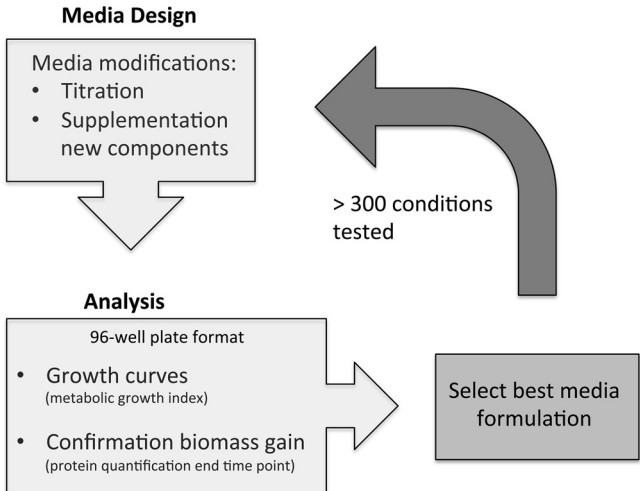

**FIG 1** Scheme showing the workflow during the process of medium optimization.

disease in pigs for which inactivated whole-cell vaccines are commercially available (28–30). Although the original vB13 composition had a poor performance for this pathogen, we were able to adapt this medium to support significant growth, providing a basis to advance in the production of *Mycoplasma* vaccines.

## RESULTS AND DISCUSSION

**Medium optimization workflow.** A culture medium for *Mycoplasma* likely requires a carbon source, amino acids, nucleotide precursors, lipids, vitamins, cofactors, essential metals, and minerals. The defined medium previously reported by Yus et al. (9) contains most of these nutrients, being capable of sustaining cellular metabolism in several *Mycoplasma* species (31, 32). However, the performance of this medium cannot support robust growth, suggesting that optimized concentrations of medium components may be crucial and that this medium probably lacks important growth factors likely present in the animal serum.

To develop a serum-free medium compatible with large-scale production, we established a workflow method to systematically test different formulations in order to optimize this medium (Fig. 1). For this, we first replaced all the amino acids, bases, vitamins, and inorganic salts present in the defined medium reported by Yus et al. (9) by RNA and the CMRL-1066 medium (Connaught Medical Research Laboratories). This commercially available medium contains most of the components described above, avoiding the addition of these components individually and therefore improving medium preparation and reproducibility. This preliminary version of the medium was referred to as vB2 (see Table S1 in the supplemental material). For optimizing medium components in vB2, we then set up a 96-well plate culture format in which growth was monitored over-time using the growth index method (9). This method estimates growth by measuring the change of absorbance (growth index = ratio 430 and 560 nm) in the culture medium and relies on the fact that *M. pneumoniae* acidifies this medium when it is metabolically active. A decrease of pH in the medium, detected by an increase in the growth index, is therefore an indirect but straightforward method to measure growth (9). To implement this method, we optimized the buffering capacity of the system, since growth in vB2 was not robust enough to change the medium pH using 100 mM HEPES, which is the buffering condition generally used in the Hayflick rich medium. For this, we tested decreased concentrations of HEPES, detecting a measurable change in the growth index with 50 mM HEPES (Fig. S1). Consequently, we used this HEPES concentration during the optimization experiments, as well as in the rich reference medium to compare yield performance. Since the buffering capacity or metabolism may change depending on the medium composition, we also measured the final protein concentration at the end of the growth curve to confirm the gain of cell biomass (Fig. 1).

**TABLE 1** Composition of different serum-free medium versions

| Component | M. pneumoniae | | | M. hyopneumoniae |
| --- | --- | --- | --- | --- |
| | vB10 | vB13 | vB13m | vH6 |
| CMRL-1066 | 0.5× | 0.5× | | 0.5× |
| L-Cysteine | | | 100 $\mu$g/mL | |
| Glucose | 7.5 g/L | 7.5 g/L | 7.5 g/L | 1.5 g/L |
| Glycerol | 0.025% | 0.025% | 0.025% | 0.01% |
| Glutamine | 2 mM | 2 mM | | 2 mM |
| PPLO | 15 mg/mL | | | |
| Yeastolate | | 10 mg/mL | 10 mg/mL | 7.5 mg/mL |
| RNA | 1 mg/mL | 1 mg/mL | | |
| Spermine | 10 $\mu$g/mL | 10 $\mu$g/mL | | 10 $\mu$g/mL |
| Thioctic acid | 0.2 $\mu$g/mL | 0.2 $\mu$g/mL | | 0.2 $\mu$g/mL |
| Pyridoxamine | 0.5 $\mu$g/mL | 0.5 $\mu$g/mL | | |
| Nicotinic acid | 0.5 $\mu$g/mL | 0.5 $\mu$g/mL | | |
| Riboflavin | 0.5 $\mu$g/mL | 0.5 $\mu$g/mL | | |
| Choline | 0.5 $\mu$g/mL | 0.5 $\mu$g/mL | | |
| Cholesterol | 33.3 $\mu$g/mL | 30 $\mu$g/mL | 30 $\mu$g/mL | 30 $\mu$g/mL |
| Phosphatidylcholine | 40 $\mu$g/mL | 40 $\mu$g/mL | 40 $\mu$g/mL | 40 $\mu$g/mL |
| Sphingomyelin | 40 $\mu$g/mL | 40 $\mu$g/mL | 40 $\mu$g/mL | 10 $\mu$g/mL |
| Palmitic acid | 16.6 $\mu$g/mL | | | |
| Oleic acid | 20 $\mu$g/mL | | | |
| Nonlipidated BSA | 0.33% | | | 0.5% |
| 2-Hydroxypropyl-$\beta$-cyclodextrin | | 5 mg/mL | 5 mg/mL | |
| Sodium pyruvate | | | | 2.2 g/L |
| Thymidine | | | | 40 $\mu$g/mL |
| Adenosine | | | | 10 $\mu$g/mL |
| Cytidine | | | | 10 $\mu$g/mL |
| Guanosine | | | | 10 $\mu$g/mL |
| Thymine | | | | 10 $\mu$g/mL |
| L-$\alpha$-Phosphatidyl-DL-glycerol | | | | 10 $\mu$g/mL |
| 1,2-Dipalmitoyl-sn-glycero-3-phosphocholine | | | | 20 $\mu$g/mL |
| HEPES | 50 mM | 50 mM | 50 mM | |
| Ampicillin | 100 $\mu$g/mL | 100 $\mu$g/mL | 100 $\mu$g/mL | 100 $\mu$g/mL |
| Phenol red (pH 7) | 0.0035% | 0.0035% | 0.0035% | 0.0035% |
| NaOH (to adjust pH to 7.7) | | | | |

Using this experimental setup, we assessed different concentrations of the original components in the performance of the vB2 preliminary medium. Additionally, we tested the contribution of other compounds that could improve performance, based on the literature and the analysis of the metabolic map of *M. pneumoniae* (9, 26). When an improvement was found, we kept it constant, and then we again tested the influence of different concentrations of the other components, as small variations in the formulation may affect their optimal range. Following this strategy, we repeated this methodology until we were unable to find a significant improvement, performing a total of more than 300 experiments and testing more than 50 different compounds.

Following this approach, we evolved the vB2 formulation to improve its yield performance. In Table S1 and Fig. S2 we present a summary of this evolution, which led to the development of two types of formulations: (i) a serum-free medium (vB10) containing bovine serum albumin (BSA) as a lipid carrier and PPLO and (ii) an animal component-free medium (vB13) in which BSA and complex animal extracts were completely removed (Table 1). Depending on the medium, these formulations provided 70 to 80% of the biomass obtained in rich medium in the 96-well-plate culture format (Table S1).

**Contribution of medium components on medium performance.** Next, we performed single-omission experiments to assess the contribution of each component in the performance of the vB13 medium (Fig. 2). As expected, the presence of CMRL medium was required for growth (Fig. 2), yet we found an optimal concentration of 0.5×, with higher concentrations negatively affecting growth. Among the other components (Table S2), CMRL provides essential amino acids, except glutamine, which was added

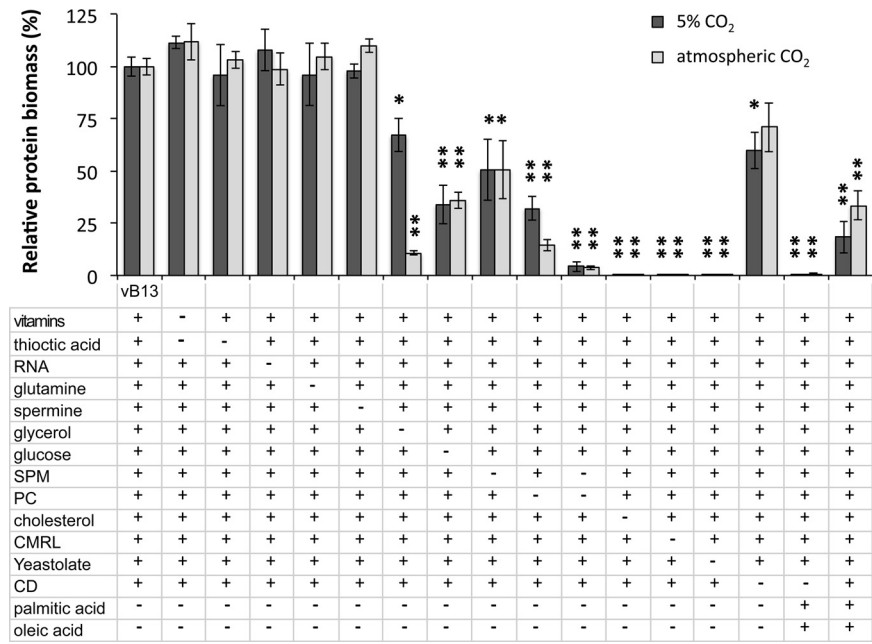

**FIG 2** Contribution of specific compounds in vB13 growth performance. Data show the percentage of protein biomass in the absence or presence of specific compounds (as indicated below the bar plot) relative to vB13 formulation. Cultures were grown in a 96-well-plate format and processed after 96 h of culture in 5% $CO_2$ or in atmospheric $CO_2$ conditions. The plus (+) symbol represents the presence of the compound, while the minus (−) symbol represents its absence. Data represent the mean ± standard error of at least 3 independent biological replicates for each medium and condition. Significance relative to the vB13 formulation was assessed by two-sided independent *t* test (*, $P < 0.05$; **, $P < 0.005$).

separately. However, the addition of glutamine or increasing amounts of different amino acids did not improve performance. Of note, *M. pneumoniae* possesses a low number of amino acid permeases and transporters, but possesses a peptide importer operon (*oppBCDF*) as well as aminopeptidases, suggesting that peptides may be an important source of amino acids (9, 13). Supporting this, the presence of peptones or similar protein hydrolysates was found to be critical for growth (Fig. 2), with increasing amounts correlating with a higher protein biomass yield (Fig. S3A). In this respect, PPLO and yeastolate showed superior performance to Bacto peptone, with the advantage that yeastolate is an animal origin-free hydrolysate (Fig. S3B).

CMRL medium also contains several essential vitamins. Supplementation of thioctic acid, which is absent in CMRL, improved growth performance during the first versions of medium development (Table S1 and Fig. S2), perhaps due to its role in oxidative defense (33). However, its positive influence was not evident in the final vB13 formulation in terms of biomass yield (Fig. 2). A similar effect was observed with the supplementation of spermine or extra amounts of vitamins such as pyridoxamine, nicotinic acid, riboflavin, and choline. On the other hand, mycoplasmas are also unable to conduct *de novo* biosynthesis of purines and pyrimidines, relying on salvage pathways and environmentally derived nucleotide precursors (34). While RNA slightly improved the growth performance at the first stages of the medium development, the supply of nucleotide precursors by CMRL, and possibly yeastolate, seemed sufficient to support growth in vB13 (Fig. 2). These results suggest that a simplified version of the vB13 formulation in the absence of vitamin supplements, amino acids, and nucleotide precursors is possible as shown below.

Glycerol derived from glycerol-containing phospholipids is likely the major source of carbon and energy for *M. pneumoniae in vivo* (35). However, under axenic culture conditions this bacterium exhibits the fastest growth in the presence of glucose (Fig. 2). Despite this, the addition of free glycerol improved growth performance, in agreement with previous results (9). While glycerol may be used as a carbon source, the growth improvement

observed may be linked to the synthesis of phospholipids and glycolipids (36). Glycerol can be converted to glycerol-3-phosphate by Glpk (MPN050) and then used as a substrate for putative acyltransferases such as PlsY (MPN350), and PlsC (MPN299) as previously suggested (9). In this scenario, the acyl groups may be delivered by the action of PlsX (MPN546), and the AcpS/P acyl carrier system encoded by MPN298 and MPN406, using fatty acids (FA) acquired from the medium. Apart from this possible role, glycerol acts as a metabolic regulator favoring pyruvate conversion to lactate by lactate dehydrogenase (LDH) with the concomitant oxidation of NADH to $NAD^+$ (37). In this scenario, activation of lactate dehydrogenase by glycerol may aid in the regeneration of the $NAD^+$ pool (37). As an example, the pyruvate dehydrogenase (PDH) complex converts pyruvate to acetyl-CoA that can be further processed to acetate, generating an extra ATP molecule after glycolysis, yet this reaction implies reduction of $NAD^+$ to NADH. In this case, glycerol may contribute to maximize ATP production without incurring NADH/$NAD^+$ redox imbalances. Intriguingly, we observed that glycerol's contribution is in fact much more significant when cells are grown under atmospheric $CO_2$ conditions than under standard growth conditions under 5% $CO_2$ (Fig. 2). Although we cannot exclude that differences in relative humidity under both culture conditions could explain these observations, it is tempting to speculate that the stimulatory effect of LDH activity by glycerol may be less effective in an environment with slightly less oxygen or that glycerol metabolism is sensitive to $CO_2$ or $O_2$ concentration. In fact, glycerol utilization in *M. pneumoniae* as a carbon source is dependent on the oxidation of glycerol 3-phosphate by the glycerol 3-phosphate oxidase GlpD (MPN051), which results in the formation of the glycolytic intermediate dihydroxyacetone phosphate and hydrogen peroxide (38).

Lipids are another essential component of the medium, consistent with the inability of mycoplasmas to synthesize both long-chain FA and cholesterol (39, 40). Although CMRL medium contains cholesterol, we found that the levels supplied are too low to support growth, requiring extra supplementation (Fig. 2). Regarding the availability of long-chain FA, we noted that the palmitic and oleic acid present in vB2 did not support robust growth, despite *M. pneumoniae* having a two-component FA kinase system consistent in two lipid transporters (*mpn472* and *mpn664*, encoding FakB and FakB2, respectively) and the FA kinase MPN547 (FakA) (41). We reasoned that FA derived from phospholipids may be a preferred source, since these are the major components of pulmonary surfactant and therefore a possible natural source of lipids (42). Supporting this, *M. pneumoniae* possesses several essential lipases (35) and a surface protein P116 (MPN213) capable of extracting and delivering cholesterol, phosphatidylcholine (PC), and sphingomyelin (SPM) (43). In fact, it has been shown that several *Mycoplasma* species, including *M. pneumoniae*, are capable of assimilating PC and SPM from the medium (39, 44–46). Similarly, the importance of SPM as a growth factor has been reported for the culture of *Spiroplasma mirum* in a defined medium (47). Accordingly, we found that the presence of both PC and SPM stimulated growth significantly (Fig. 2), suggesting the two phospholipids play different roles (27). Strikingly, we did not detect significant improvements when using other types of lipids such as cardiolipin or ceramide. We also tested the possibility of supplying palmitic and oleic acid using Tween 40 and Tween 80, respectively (48). Although they were capable of substituting the free FA at certain concentrations, all these compounds appeared to be toxic in the absence of BSA (Table S1 and Fig. S2).

How lipids are delivered to the cell in an assimilable and nontoxic manner is also an important factor contributing to growth performance. As previously reported for other serum-free media (49), delipidated BSA was capable of replacing animal serum to some extent when combined with lipids such as cholesterol or FA in the vB10 formulation (Table S1 and Fig. S2). In order to remove animal components from the vB10 medium, PPLO was substituted with yeastolate and BSA, with cyclodextrin, resulting in the version vB13 animal component-free medium. Hydroxypropyl-$\beta$-cyclodextrin was previously shown to be a suitable carrier of cholesterol and FA for *Mycoplasma capricolum* culture (50). We found that hydroxypropyl-$\beta$-cyclodextrin promoted growth of *M. pneumoniae* in the absence of serum and BSA, yet it did not completely prevent the toxicity of free

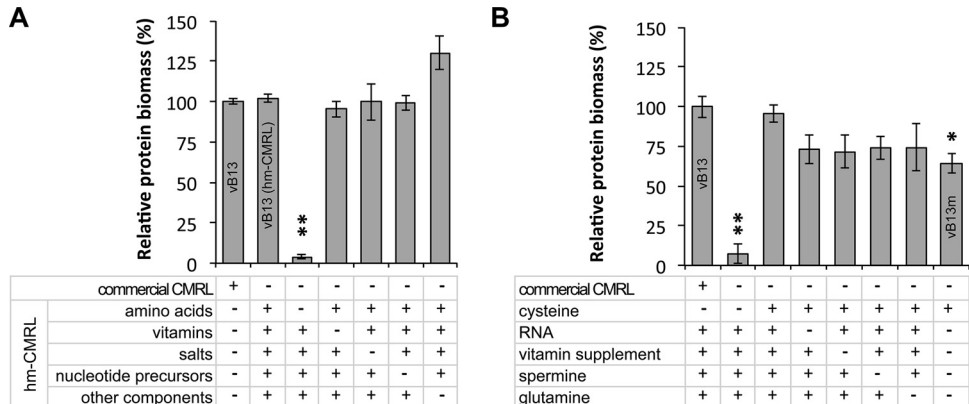

**FIG 3** Essential culture components required for *M. pneumoniae* growth. (A) Identification of groups of compounds in CMRL that are required for supporting growth in vB13. Data show the percentage of protein biomass relative to vB13 of different medium versions, in which commercial CMRL has been substituted with a full home-made CMRL version (hm-CMRL) or variants lacking specific groups of CMRL compounds. Cultures were grown in a 96-well-plate format and processed after 96 h of culture. The plus (+) and minus (−) symbols represent the presence or absence, respectively, of the indicated group of compounds (see Table S2 for a detailed list). Data represent the mean ± standard error of 3 independent biological replicates. Significance relative to the vB13 formulation was assessed by two-sided independent *t* test (*, $P < 0.05$; **, $P < 0.005$). (B) Identification of minimal medium constituents (vB13m) required for supporting *M. pneumoniae* culture. Data show the percentage of protein biomass relative to vB13 as described in panel A. Components that are common among the tested medium formulations are not shown for clarity purposes. The detailed formulation of vB13m can be found in Table 1.

FA (Fig. 2). Based on these results and the fact that FA are provided by phospholipids, we removed palmitic and oleic acid from the vB13 formulation (Table 1).

**Essential CMRL components in the serum-free medium.** The commercially available CMRL-1066 medium was used during medium development to facilitate medium preparation and provide several essential components that would otherwise be supplemented separately. To identify which of the CMRL components were strictly necessary for growth, we performed omission experiments of specific CMRL compounds. For this, we prepared a home-made CMRL (hm-CMRL) medium by adding each component individually and a set of CMRL variants lacking a specific group of compounds, namely, amino acids, vitamins, inorganic salts, nucleotide precursors, and other components (listed in Table S2). As expected, substitution of the commercial CMRL present in vB13 with the hm-CMRL resulted in a similar growth performance (Fig. 3A). Except for the CMRL variant lacking amino acids, all the other deficient variants also supported growth (Fig. 3A), suggesting that one, or several, amino acids in CMRL could not be replaced as an active compound by the protein hydrolysate (yeastolate). To identify the essential amino acid, we tested hm-CMRL variants containing groups of only 4 amino acids. Of the 5 groups tested, only one was capable of restoring growth (Fig. S4A). Then, we separately tested each of the 4 amino acids constituting this group, uncovering that addition of cysteine, or its disulfide form cystine, was sufficient to restore growth in the absence of CMRL (Fig. 3B and Fig. S4B). This observation was consistent with our previous studies showing that protein hydrolysates such as peptones cannot efficiently substitute for cysteine as a free amino acid (9). In addition, these results demonstrated that vitamins and nucleotide precursors supplied by CMRL were not functioning as essential active components. In fact, removal of RNA and the vitamin supplement from vB13 did not significantly impair growth of a vB13 medium version in which CMRL was substituted with cysteine (Fig. 3B), indicating that yeastolate is an efficient source of vitamins and nucleotide precursors. Attempts to substitute yeastolate from vB13 by increasing the amounts of free amino acids (up to 2 mM) were unsuccessful, suggesting that peptides are a preferred source of amino acids. In fact, the peptide importer operon (*oppBCDF*) is essential in *M. pneumoniae*, suggesting that this bacterium is limited on amino acid transporters and depends on the peptide

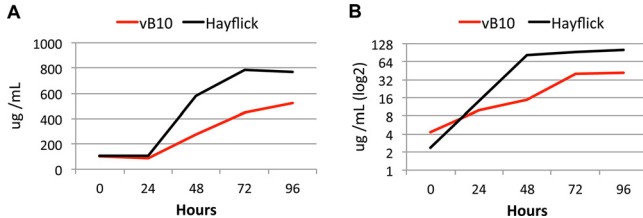

**FIG 4** Growth curve analyses of *M. pneumoniae* grown in vB10 (serum-free). (A and B) Cell growth in vB10 along with 96 h of culture in 25-cm² tissue culture flasks containing 5 mL of medium was assessed by (A) protein and (B) DNA biomass quantification and compared to rich medium (Hayflick). Growth performance in vB13 is described in reference 27.

import system (51). Nevertheless, we cannot exclude that yeastolate may also contain other undefined essential growth factors. Overall, these results suggest minimum component requirements to create custom medium preparations to support the growth of *M. pneumoniae* (see vB13m in Table 1 and Fig. 3B).

**Scaleup and assessment of long-term passaging.** To confirm the growth performance obtained during the screening process in the 96-well-plate culture format, we scaled up the culture conditions to 5 mL of medium using 25-cm² tissue culture flasks. Cultures were started with a 1:200 dilution of inoculum, and growth was monitored over time by measuring both the protein and DNA biomass during 96 h of culture. The two quantification methods showed that growth performance was very similar in the vB10 (Fig. 4) and vB13 (27) medium versions, reaching 60 to 70% of the biomass obtained in rich medium. Growth was also successful using higher volumes of medium using 75-cm² tissue culture flasks.

Next, we tested whether vB10 and vB13 formulations supported long-term growth. Cells were successfully grown in the two media by performing serial passaging at a ratio of 1:25 every 3 to 5 days during the 10 consecutive passages that were attempted. Although vB13 enabled long-term growth, the subculturing process was less efficient and reproducible than that of vB10, requiring longer cultivation periods, especially during the last passages. Therefore, the vB10 formulation containing BSA exhibited better performance in long-term culture.

**Gene expression analysis under vB13 medium growth conditions.** To examine the physiological effect of growing *M. pneumoniae* in the absence of animal components, we compared the gene expression of cells grown in vB13 or rich medium, using both genome-wide transcriptomic and proteomic analysis. In general, the patterns of gene expression were very similar in the two media, showing a strong Pearson's correlation for both transcript ($r = 0.97$) and protein ($r = 0.89$) levels (Fig. 5). At the transcriptional level, we found 6 and 20 protein-coding genes significantly up- or downregulated, respectively, showing a fold change greater than 1 $\log_2$ (Table S3 and Fig. S5A). Differences in gene expression were larger at the proteomic levels, but not statistically significant (Table S4 and Fig. S5B). Although quality assessment of the final bacterial product is required, these results show a minor impact in gene expression, supporting vB13 medium as a suitable and potential formulation for the production of *M. pneumoniae*-based therapies.

**Serum-free medium adaptation for *M. hyopneumoniae* growth.** We wondered whether this medium formulation could be universal for *Mycoplasma* growth or of nutritional requirements might differ among different species. To test this, we assessed the performance of vB13 to grow *M. hyopneumoniae*, for which inactivated whole-cell vaccines are available (28–30). We found that the vB13 composition did not support growth of this swine pathogen (Fig. 6A). Therefore, we followed the strategy described above to systematically test different concentrations and compounds to adapt the vB13 composition for *M. hyopneumoniae*. Since this pathogen grows in suspension, we performed the growth screening directly in 4-mL cultures and monitored growth by measuring DNA biomass at 96 h of culture (see Materials and Methods). Although *M. hyopneumoniae* can ferment glucose, its metabolism produces less lactic acid than *M. pneumoniae*.

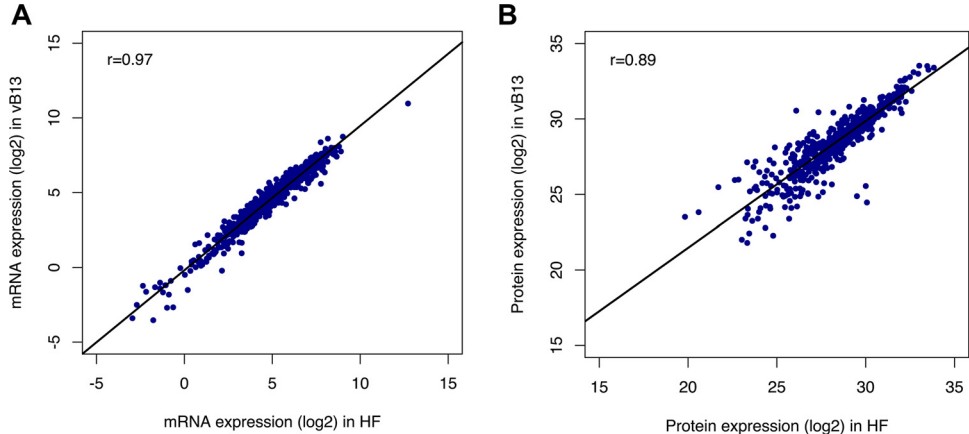

**FIG 5** Pearson's correlation analysis of gene expression of *M. pneumoniae* grown in rich medium (HF) or animal component-free medium (vB13). (A) Comparative analyses of mRNA levels (FPKM) assessed by RNA-seq experiments. (B) Comparative analyses of protein levels assessed by mass spectrometry experiments. RNA-seq and proteomic data are represented as the mean of two biological replicates.

Consequently, we removed HEPES, allowing us to assess qualitative growth by monitoring color change.

Among the modifications that were found to improve *M. hyopneumoniae* growth performance (Fig. 6A) was the addition of sodium pyruvate with a concomitant reduction of the glucose supply as previously described (52). We also found that glycerol amounts above 0.01% resulted in nonreproducible cultures. Furthermore, the metabolic map of *M. hyopneumoniae* lacks the thymidylate synthetase gene (*thyA*), suggesting that synthesis of dTMP may depend on the activity of the thymidine kinase (*tdk*) (53). In fact, disruption of *thyA* in *M. pneumoniae* results in upregulation of *tdk*, indicating that thymidine uptake

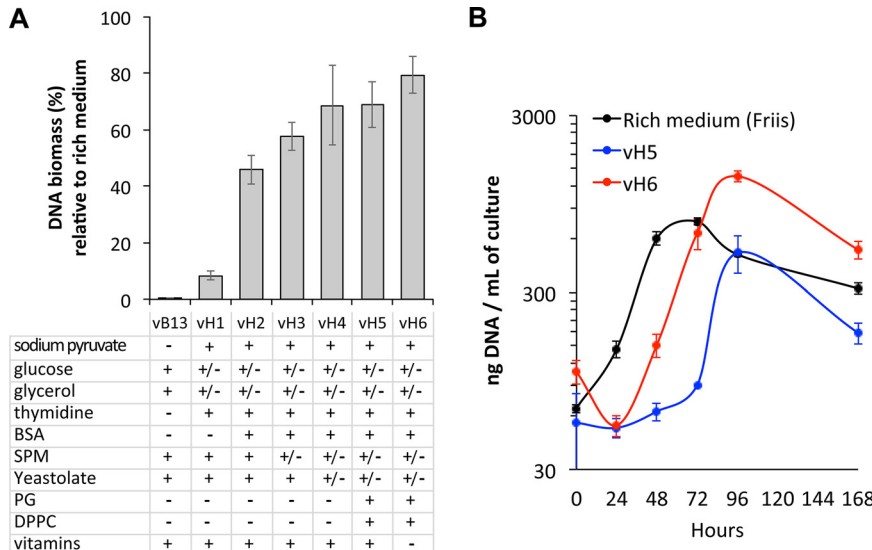

**FIG 6** Serum-free medium adaptation for *M. hyopneumoniae* growth. (A) Growth performance of different medium versions obtained during the optimization process of the vB13 formulation for *M. hyopneumoniae* growth adaptation. Data show the percentage of DNA biomass yield relative to that of rich medium (modified Friis medium) after 96 h in 4-mL cultures. Data represent the mean ± standard error of a minimum of 3 biological replicates except for vB13 (*n* = 2). Below the bar plot are shown the main contributing factors of medium adaptation. The plus (+) symbol represents the presence of the compound, the minus (−) symbol represents its absence, while +/− indicates a decrease in the concentration of the compound compared to vB13. The detailed formulation of each medium version is shown in Table S5. (B) Growth curve analysis of *M. hyopneumoniae* in rich medium and serum-free medium vH5 and vH6. Cell growth was monitored over time in 30-mL cultures by DNA biomass quantification. Data represent the mean ± standard error of 3 biological replicates for each medium.

coupled with Tdk activity may compensate for the lack of ThyA activity (54). Since RNA apparently did not provide any growth advantage, we replaced it with nucleosides, including increased thymidine amounts. A major difference compared to *M. pneumoniae* was the inability of cyclodextrin to support efficient growth of *M. hyopneumoniae* in the absence of BSA (Fig. 6A). Similarly, *M. hyopneumoniae* seemed to tolerate less SPM than *M. pneumoniae*. Although its presence was beneficial, SPM amounts above 10 $\mu$g/mL resulted in growth inhibition. PC instead seems to play a key role as a growth factor. Other positive modifications included the reduction of the yeastolate concentration, and the addition of phospholipids such as L-$\alpha$-phosphatidyl-DL-glycerol (PG) and 1,2-dipalmitoyl-sn-glycero-3-phosphocholine (DPPC), which improved culture consistency. Finally, we found that the addition of extra vitamins impaired growth (Fig. 6A). Overall, our optimization process resulted in medium formulations reaching 60 to 80% of the DNA biomass obtained with rich medium (Fig. 6A and Table S5).

To further assess growth performance, we scaled up the growth conditions to 30 mL and performed growth curve analyses by measuring DNA biomass over time. As shown in Fig. 6B, we reproduced the enhanced performance of vH6 compared to vH5, which promoted growth with shorter lag phases, approximating the yields obtained in rich medium after 72 h of growth. Based on these results, we conclude that medium optimization is required for the particular metabolic activity associated with each species and that the vH6 formulation can be used for *M. hyopneumoniae* vaccine production upon the process of industrial optimization.

**Conclusion.** In this study, we developed a serum-free medium capable of supporting robust growth of *M. pneumoniae*. We provide several medium formulations that can be chosen depending on the application, including a medium composition free of animal components. We showed that depending on the *Mycoplasma* species, further optimization might be required, as illustrated by the cultivation of *M. hyopneumoniae*, an important swine pathogen with available vaccines. The results presented in this study could assist in the large-scale production of *Mycoplasma*-based therapies in the absence of animal components upon industrial optimization, contributing to the development of safer and cheaper products.

## MATERIALS AND METHODS

**Bacterial strains and preparation of inocula.** Wild-type *M. pneumoniae* strain M129 and the nonpathogenic strain J of *M. hyopneumoniae* were cultivated in different serum-free medium formulations and compared to modified Hayflick (9) and Friis (52) control rich media, respectively. To avoid variability in the culture tests, we prepared a large working stock of inocula for each species as follows.

For *M. pneumoniae*, culture flasks of 300 cm$^2$ containing 75 mL of Hayflick medium were inoculated with 200 $\mu$L of bacterial cell suspension and cultured at 37°C and 5% CO$_2$. After 72 h of culture, the culture medium was removed, and cells were scraped from the flask into 10 mL of fresh medium. Then, the bacterial cell suspension was stored at $-80$°C in aliquots of 200 $\mu$L until needed.

For *M. hyopneumoniae*, 50-mL conical tubes containing 10 mL of Friis medium were inoculated with 250 $\mu$L of bacterial cell suspension and cultured at 37°C in an orbital shaker set at 180 rpm. After 72 h of growth, the 10-mL culture was diluted in 90 mL of fresh medium in a 250-mL Erlenmeyer flask. After 24 h of culture, the 100 mL of culture was diluted again by adding 150 mL of fresh medium in a 500-mL Erlenmeyer and cultured for 24 h more. Then, bacterial cells were harvested by centrifugation (14,100 $\times$ *g*, 10 min), and the pellet was resuspended in 30 mL of fresh medium. Finally, the cell suspension was stored at $-80$°C in aliquots of 400 $\mu$L until needed.

For each experiment, an aliquot was thawed to avoid repeating freezing and thawing cycles. To measure the inoculum concentration, one aliquot was used to quantify the protein biomass using the Pierce bicinchoninic acid (BCA) protein assay kit.

**Culture conditions and growth assessment during medium development.** To optimize the formulation of the serum-free medium in *M. pneumoniae*, we established a workflow method to monitor the growth of multiple medium formulations in a 96-well-plate format incubated in a Tecan Spark plate reader at 37°C and atmospheric CO$_2$ conditions. A duplicate plate was cultured in parallel under standard growth conditions using a cell culture incubator set at 37°C and 5% CO$_2$. Each medium was tested at least in duplicate wells containing 200 $\mu$L of medium inoculated with 1.5 $\mu$g of a frozen stock as starting inocula (approximately 1:100 dilution). Medium performance was monitored by growth curve analyses based on the growth index method previously described (9), in which growth is estimated over time by measuring the change of absorbance (ratio of 430 and 560 nm). To confirm the gain of cell biomass, we also measured the final protein concentration at the end of the growth curve (typically 4 to 5 days) for each duplicate plate. For this, wells were washed twice with phosphate-buffered saline (PBS), and attached cells were lysed with 100 $\mu$L of lysis buffer (10 mM Tris, pH 8, 6 mM MgCl$_2$, 1 mM EDTA,

100 mM NaCl, and 0.1% Triton X-100 plus a cocktail of protease inhibitors). Then, the protein biomass for each well lysate was measured in duplicate using the Pierce BCA protein assay kit.

Culture conditions in the 96-well-plate format were not optimal for growing *M. hyopneumoniae*, even using shaking conditions. Determination of cell biomass gain by protein quantification also resulted in high background signals, likely due the nonadherent nature of this bacterium. Consequently, we set up the following culture conditions to estimate growth and aid the medium optimization process. For each experiment, 3 culture replicates of 4 mL each containing the tested medium or Friis medium were inoculated with frozen stocks (1:200 dilutions). Cultures were incubated at 37°C in an orbital shaker (180 rpm) in 15-mL conical tubes, and the total DNA biomass was estimated after 96 h of culture as follows. The 4-mL bacterial culture was harvested by centrifugation ($14,100 \times g$, 10 min), and the pellet was directly lysed with 300 $\mu$L of tissue and cell lysis solution plus proteinase K from the MasterPure DNA purification kit (Lucigen). DNA was extracted following the recommendations of the kit manufacturer, and DNA was measured using a fluorometric method using the Qubit double-stranded DNA (dsDNA) high-sensitivity (HS) assay kit (Invitrogen).

The formulation of different medium versions for *M. pneumoniae* and *M. hyopneumoniae* is detailed in Table S1 and Table S5, respectively.

**Scale-up and growth curve analysis.** Growth curve analysis of vB10 and vB13 medium formulations was performed in larger volume cultures for *M. pneumoniae* and compared to Hayflick rich medium as follows. Several tissue culture flasks of 25 cm² containing 5 mL of medium were inoculated with frozen stocks (1:200 dilution) and incubated at 37°C and 5% $CO_2$. Protein and DNA biomass were then measured at different time points (0 h, 24 h, 48 h, 72 h, 96 h) as follows. For each time point, cells were scraped from the flasks in the culture medium, and 1 mL of cell suspension was harvested by centrifugation ($13,100 \times g$, 10 min). This was performed in duplicate to obtain samples for both protein and DNA measurements. For protein biomass quantification, the cell pellet was washed twice with PBS and lysed with 100 $\mu$L of lysis buffer (10 mM Tris, pH 8, 6 mM $MgCl_2$, 1 mM EDTA, 100 mM NaCl, and 0.1% Triton X-100 plus a cocktail of protease inhibitors) prior to duplicate protein measurements using the Pierce BCA protein assay kit. For DNA biomass quantification, the cell pellet was directly lysed, and the DNA was extracted using the MasterPure DNA purification kit (Lucigen) following the recommendations of the kit manufacturer. Finally, extracted DNA for each time point was measured using a fluorometric method using the Qubit dsDNA HS assay kit (Invitrogen).

To evaluate in more detail the performance of certain medium formulations for *M. hyopneumoniae*, 125-mL Erlenmeyer flasks containing 30 mL of medium were inoculated in triplicate with frozen stocks (1:200 dilutions) and incubated at 37°C in an orbital shaker set at 180 rpm. Then, 1 mL of culture sample was collected at different time points (0 h, 24 h, 48 h, 72 h, 96 h, 168 h), and cells were harvested by centrifugation ($14,100 \times g$, 10 min) prior to DNA quantification following the methods described above.

**Passaging experiments.** To test whether the vB10 and vB13 medium formulations could support growth after consecutive passages, we performed passaging experiments as follows. Cultures in tissue culture flasks of 25-cm² containing 5 mL of medium were grown for 3 to 5 days. Then, cells were scraped from the flasks into the same culture medium and 200 $\mu$L of cell suspension diluted in 5 mL of fresh medium to start a new culture. This process was repeated for up to 10 consecutive passages.

**Sample preparation and RNA-seq analysis.** *M. pneumoniae* was grown in duplicate in tissue culture flasks containing 5 mL of Hayflick medium or animal component-free medium vB13. After 72 h of culture at 37°C and 5% $CO_2$, the culture medium was changed with fresh medium, and the cells were further incubated for 6 h prior to RNA extraction. RNA was isolated using the miRNeasy kit (Qiagen) following the manufacturer's instructions, including the in-column DNase I treatment. The quality of RNA (amount and integrity) was assessed using a BioAnalyzer device (Agilent), and the transcriptome sequencing (RNA-seq) libraries were prepared at the CRG Genomics Unit using the TruSeq stranded mRNA sample prep kit v2 (55). Sequencing was performed using a HiSeq 2500 instrument (Illumina) with HiSeq v4 chemistry and $2 \times 50$-bp paired-end reads.

Processing of sequencing reads was performed as described (55) using the wild-type reference genome of *M. pneumoniae* M129 (NCBI accession no. NC_000912.1). Differential expression analysis was performed using edgeR v3.12.1 (56–59) with trimmed mean of M-values (TMM) normalization and classical pairwise comparison between serum-free and rich medium conditions. Multiple-test correction was applied using the Benjamini-Hochberg method (60), and fold changes greater than 1 $\log_2$ with a corrected $q$ value smaller than 0.05 were considered significant (Table S3).

**Sample preparation and mass spectrometry analysis.** *M. pneumoniae* was grown in duplicate in tissue culture flasks of 75 cm² containing 20 mL of Hayflick medium or animal component-free medium vB13. After 72 h of culture at 37°C and 5% $CO_2$, the culture medium was changed with fresh medium, and the cells were further incubated for 6 h. At this point, cells were washed twice with PBS, scraped from the flasks, and centrifuged at $13,100 \times g$ for 10 min. The pellet was resuspended in lysis buffer (4% SDS, 100 mM HEPES, pH 7.4) and disrupted using a Bioruptor sonication system (Diagenode) with an on/off interval time of 30/30 s at high frequency for 5 min. Cell lysates were spun, and the extracted protein was quantified using the Pierce BCA protein assay kit. Finally, samples were digested with trypsin using the filter-aided protocol (FASP), and they were analyzed using an LTQ-Orbitrap Velos Pro mass spectrometer (Thermo Fisher Scientific, San Jose, CA, USA) coupled to an EASY-nLC 1000 (Thermo Fisher Scientific [Proxeon], Odense, Denmark). Peptides (1 $\mu$g) were loaded onto the 2-cm NanoTrap column with an inner diameter of 100 $\mu$m packed with $C_{18}$ particles of 5-$\mu$m particle size (Thermo Fisher Scientific) and were separated by reversed-phase chromatography using a 25-cm column with an inner diameter of 75 $\mu$m, packed with 1.9-$\mu$m $C_{18}$ particles (Nikkyo Technos Co., Ltd. Japan). Chromatographic gradients started at 93% buffer A (0.1% formic acid in water) and 7% buffer (0.1% formic acid in acetonitrile) B with a flow rate of 250 nL/min for 5 min and gradually increased to 65%

buffer A and 35% buffer B in 120 min. After each analysis, the column was washed for 15 min with 10% buffer A and 90% buffer B.

The mass spectrometer was operated in positive ionization mode with nanospray voltage set at 2.1 kV and source temperature at 300°C. Ultramark 1621 was used for external calibration of the Fourier Transform (FT) mass analyzer prior to the analyses, and an internal calibration was performed using the background polysiloxane ion signal at $m/z$ 445.1200. The acquisition was performed in data-dependent acquisition (DDA) mode, and full mass spectrometry (MS) scans with 1 microscan at a resolution of 60,000 were used over a mass range of $m/z$ 350 to 2,000 with detection in the Orbitrap. Auto gain control (AGC) was set to 1E6, and dynamic exclusion (60 s) and charge state filtering disqualifying singly charged peptides were activated. In each cycle of DDA analysis, following each survey scan, the top 20 most intense ions with multiple-charge ions above a threshold ion count of 5,000 were selected for fragmentation. Fragment ion spectra were produced via collision-induced dissociation (CID) at a normalized collision energy of 35%, and they were acquired in the ion trap mass analyzer. AGC was set to 1E4, and an isolation window of 2.0 $m/z$, an activation time of 10 ms, and a maximum injection time of 100 ms were used. All data were acquired with Xcalibur software v2.2. Digested bovine serum albumin (New England Biolabs) was analyzed between each sample to avoid sample carryover and to ensure stability of the instrument. QCloud (61) was used to control instrument longitudinal performance during the project.

Acquired spectra were analyzed using the Proteome Discoverer software suite (v2.0, Thermo Fisher Scientific) and the Mascot search engine (v2.5, Matrix Science) (62). The data were searched against a home-made database consisting of a list of common contaminants and all possible *M. pneumoniae* open reading frames (ORFs) of >19 amino acids (aa; 87,051 entries). For peptide identification, a precursor ion mass tolerance of 7 ppm was used for the MS1 level, trypsin was chosen as the enzyme, and up to three missed cleavages were allowed. The fragment ion mass tolerance was set to 0.5 Da for MS2 spectra. Oxidation of methionine and N-terminal protein acetylation were used as variable modifications, whereas carbamidomethylation on cysteines was set as a fixed modification. The false-discovery rate (FDR) in peptide identification was set to a maximum of 5%. Peptide quantification data were retrieved from the "precursor ion area detector" node of Proteome Discoverer (v2.0) using 2 ppm mass tolerance for the peptide extracted ion current (XIC). The obtained values were used to estimate the protein abundances with the average of the three most intense unique peptides per protein. Fold changes ($log_2$) were calculated for those proteins detected under both growth conditions, and a two-tailed $t$ test assuming unequal variances was performed when signal was detected in all replicates. Multiple-test correction was applied using the Benjamini-Hochberg method (60), and fold changes with a corrected $q$ value smaller than 0.05 were considered significant (Table S4).

**Data availability.** The RNA sequencing data have been deposited in the ArrayExpress repository with the data set identifiers E-MTAB-10111 and E-MTAB-12308 for samples grown in Hayflick and vB13 animal component-free medium, respectively. The mass spectrometry proteomics data have been deposited in the ProteomeXchange Consortium via the PRIDE (63) partner repository with the data set identifier PXD037431.

## SUPPLEMENTAL MATERIAL

Supplemental material is available online only.
**SUPPLEMENTAL FILE 1**, PDF file, 1.3 MB.
**SUPPLEMENTAL FILE 2**, XLS file, 0.2 MB.
**SUPPLEMENTAL FILE 3**, XLS file, 0.2 MB.

## ACKNOWLEDGMENTS

This work has been supported by the European Union's Horizon 2020 research and innovation program under grant agreement 634942 (MycoSynVac), the European Research Council (ERC) under the European Union's Horizon 2020 research and innovation program under grant agreement 670216 (MYCOCHASSIS), and Conveni La Caixa 2020-2023 (LA CAIXA-EGA, LCF/PR/GN13/10260009). We also acknowledge the support of the Spanish Ministry of Science and Innovation to the EMBL partnership, the Centro de Excelencia Severo Ochoa, and the CERCA Program from the Generalitat de Catalunya.

We also acknowledge the staff of the CRG Genomics Unit for performing RNA-seq library preparation and sequencing. The proteomics analyses were performed in the CRG/UPF Proteomics Unit, which is part of the Proteored, PRB3, and is supported by grant PT17/0019 of the PE I+D+i 2013-2016, funded by ISCIII and ERDF.

We also thank Jetta Bijlsma and Tjerko Kamminga for sharing the nonpathogenic J strain of *M. hyopneumoniae*.

Patent applications were filed for the *M. pneumoniae* (WO2021078935) and *M. hyopneumoniae* (WO2021078938) serum-free media. Luis Serrano and Maria Lluch-Senar are cofounders of Pulmobiotics Ltd., and Maria Lluch-Senar is currently an employee of this company.

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
