## [Reviewer comments · Microbiology Spectrum]

Microbiology Spectrum

Development of a serum-free medium to aid large-scale production of Mycoplasma-based therapies

Raul Burgos, Eva Garcia-Ramallo, Daniel Shaw, Maria LLuch-Senar, and Luis Serrano

Corresponding Author(s): Raul Burgos, Centre for Genomic Regulation (CRG). The Barcelona Institute of Science and Technology

Review Timeline:

Submission Date:	November 25, 2022
Editorial Decision:	January 29, 2023
Revision Received:	March 24, 2023
Accepted:	April 3, 2023

Editor: Jing Han

Reviewer(s): Disclosure of reviewer identity is with reference to reviewer comments included in decision letter(s). The following individuals involved in review of your submission have agreed to reveal their identity: Robert Doug Hardy (Reviewer #1); Mingqiang Li (Reviewer #2)

Transaction Report:

DOI: <https://doi.org/10.1128/spectrum.04859-22>

January 29, 2023

Dr. Raul Burgos
Centre for Genomic Regulation (CRG). The Barcelona Institute of Science and Technology
Dr Aiguader 88
Barcelona, Barcelona 08003
Spain

Re: Spectrum04859-22 (Development of a serum-free medium to aid large-scale production of Mycoplasma-based therapies)

Dear Dr. Raul Burgos:

Link Not Available

Sincerely,

Jing Han

Journals Department
Reviewer comments:

Reviewer #1 (Comments for the Author):

I enjoyed reading this fundamental manuscript on mycoplasma. The results of this manuscript are very useful for expediting mycoplasma research, as well as better understanding mycoplasma metabolism. Well done.

Reviewer #2 (Comments for the Author):

This research article developed a culture medium free of animal serum and other animal components for mycoplasma pneumonia growth, assisting in the advancement of large-scale production of safe mycoplasma vaccines. In addition, this study

established a workflow method to systematically test different compounds and concentrations, resulting in optimized formulations capable of supporting serial passaging and robust growth. The author did present some data to prove his conclusion. However, there are some questions, editing, and formatting errors in the manuscript needed to be addressed. Overall, this article needs further revision before publication.

Comments:

- (1) Page 2, line 33: the abbreviations of "CMRL" should be written in their full names when they first appear.
- (2) Page 2, line 35, line 36, line 37, and line 38: I don't think it's necessary to capitalize the first letter of these words (such as Cysteine, Yeastolate, Cysteine...). Please check the full text.
- (3) Page 2, line 43: The sentence "Diseases associated with Mycoplasmas are an important economic burden in both human and livestock systems" needs to be refined
- (4) Page 4, line 69: the abbreviations of "M. pneumoniae" should be written in their full names when they first appear.
- (5) Page 4, line 77: "BHI (brain heart infusion broth), PPLO (beef heart infusion and peptones)" change to "brain heart infusion broth (BHI), beef heart infusion and peptones (PPLO)"
- (6) Page 6, line7, line120: "lipids, vitamins and co-factors, essential metals and" change to "lipids, vitamins, co-factors, essential metals and...."
- (7) Page 11, line 302, line 305; page 13 line 366: "96h", "75cm²" and "72h" change to "96 h", "75 cm²" and "72 h". It needs a space before the unit. Please check the full text.
- (8) In the part of REFERENCES: Please uniform reference journal names: full or abbreviated? Such as "Physiological Reviews", "Mol Syst Biol.". In addition, if the journal name is presented as an abbreviation, add a dot after each abbreviation word, such as "Mol Syst Biol." Change to "Mol. Syst. Biol."

Other questions:

- (1) Page 6, line129-131: "To facilitate medium preparation and reproducibility, we replaced amino acids, bases, vitamins and inorganic salts by RNA and the commercially available CMRL-1066 medium." What is the basis of this change, and do not see the relevant references.
- (2) Page 6, line 135: "This method estimates growth by measuring the change of absorbance (ratio 430 and 560 nm).....". What is the principle of this method, and did not see the reference
- (3) Page 6, line 141-142: "we tested decreased concentrations of HEPES, detecting a measurable change in pH with 50mM HEPES (Fig. S1)." The data did not show the changes in pH indicators?
- (4) Page 6, line 141-144: The pH indicators in different mediums should change during the culture. You should judge the effect of different mediums by the degree of pH change, not simply by whether the pH index has changed.
- (5) Page 7, line 151: ".....and the analysis of the metabolic map of M.pneumoniae....." Where is this data?
- (6) Page 12, line 323-325: "these results show a minor impact in gene expression....." Whether this change will also affect the function of subsequent vaccines
- (7) In this study, a serum-free medium was developed to replace the traditional medium. In the optimization process, whether there are still some natural ingredients in serum cannot be determined by artificial optimization?

Staff Comments:

Preparing Revision Guidelines

Please return the manuscript within 60 days; if you cannot complete the modification within this time period, please contact me. If you do not wish to modify the manuscript and prefer to submit it to another journal, please notify me of your decision immediately so that the manuscript may be formally withdrawn from consideration by Microbiology Spectrum.

If your manuscript is accepted for publication, you will be contacted separately about payment when the proofs are issued;

please follow the instructions in that e-mail. Arrangements for payment must be made before your article is published. For a complete list of **Publication Fees**, including supplemental material costs, please visit our website.

Responses to reviewer comments

Reviewer #1 (Comments for the Author):

I enjoyed reading this fundamental manuscript on mycoplasma. The results of this manuscript are very useful for expediting mycoplasma research, as well as better understanding mycoplasma metabolism. Well done.

We thank reviewer #1 for the reviewing process and for the positive assessment of our manuscript.

Reviewer #2 (Comments for the Author):

This research article developed a culture medium free of animal serum and other animal components for mycoplasma pneumonia growth, assisting in the advancement of large-scale production of safe mycoplasma vaccines. In addition, this study established a workflow method to systematically test different compounds and concentrations, resulting in optimized formulations capable of supporting serial passaging and robust growth. The author did present some data to prove his conclusion. However, there are some questions, editing, and formatting errors in the manuscript needed to be addressed. Overall, this article needs further revision before publication.

We thank reviewer #2 for carefully reading the manuscript and providing constructive feedback to improve our manuscript. Please, see below a detailed point-by-point response to the specific comments and questions.

Comments:

(1) Page 2, line 33: the abbreviations of "CMRL" should be written in their full names when they first appear.

Following the editorial guideline, we tried to avoid abbreviations in the abstract. Instead, we added the abbreviation of CMRL (Connaught Medical Research Laboratories) in page 6, line 131, when it first appears in the introduction.

(2) Page 2, line 35, line 36, line 37, and line 38: I don't think it's necessary to

capitalize the first letter of these words (such as Cysteine, Yeastolate, Cysteine...).
Please check the full text.

Thank you for the comment. We have corrected this as suggested.

(3) Page 2, line 43: The sentence "Diseases associated with Mycoplasmas are an important economic burden in both human and livestock systems" needs to be refined

We have modified this sentence as follows: "Mycoplasma infections have a significant negative impact on both livestock production and human health".

(4) Page 4, line 69: the abbreviations of "M. pneumoniae" should be written in their full names when they first appear.

The full name of "Mycoplasma pneumoniae" is defined for the first time in page 4, line 62.

(5) Page 4, line 77: "BHI (brain heart infusion broth), PPLO (beef heart infusion and peptones)" change to "brain heart infusion broth (BHI), beef heart infusion and peptones (PPLO)"

Thank you for noting this. We have modified the sentence as suggested.

(6) Page 6, line7, line120: "lipids, vitamins and co-factors, essential metals and" change to "lipids, vitamins, co-factors, essential metals and...."

The sentence has been changed as suggested.

(7) Page 11, line 302, line 305; page 13 line 366: "96h", "75cm2" and "72h" change to "96 h", "75 cm2" and "72 h". It needs a space before the unit. Please check the full text.

Thank you for noting this error. We have now corrected this through the manuscript.

(8) In the part of REFERENCES: Please uniform reference journal names: full or abbreviated? Such as "Physiological Reviews", "Mol Syst Biol.". In addition, If the journal name is presented as an abbreviation, add a dot after each abbreviation word, such as "Mol Syst Biol." Change to "Mol. Syst. Biol."

The format of the reference section has been corrected following the journal editorial style.

Other questions:

(1) Page 6, line 129-131: "To facilitate medium preparation and reproducibility, we replaced amino acids, bases, vitamins and inorganic salts by RNA and the commercially available CMRL-1066 medium." What is the basis of this change, and do not see the relevant references.

We used as a starting formulation the defined medium previously developed in our laboratory (reference Yus et al., 2009). This medium is not capable of supporting robust growth (as stated in page 6, line 121-126) and contains many components, such as amino acids, individual bases, vitamins or inorganic salts that are added separately. This makes the preparation of the media difficult and can result in lack of reproducibility when it is prepared. Since RNA (to substitute nucleotide precursors) and the CMRL medium contain most of these components (in addition to other co-factors), we decided to use RNA and this commercial medium to substitute many of the components that were added separately in the previous medium. This strategy facilitated the preparation of the medium and its reproducibility, especially when different media formulations are tested. To make this clear, the sentence in page 6, lines 129-131 has been modified as follows (now page 6, lines 129-135):

"For this, we first replaced all the amino acids, bases, vitamins and inorganic salts present in the defined medium reported by Yus et al., (9), by RNA and the CMRL-1066 medium (Connaught Medical Research Laboratories). This commercially available medium contains most of the components described above, avoiding the addition of these components individually, and therefore improving medium preparation and reproducibility".

(2) Page 6, line 135: "This method estimates growth by measuring the change of absorbance (ratio 430 and 560 nm).....". What is the principle of this method, and did not see the reference

As stated in page 6, line 134 (now page 6, line 137), the "growth index" method was previously described by Yus et al (2009) and it has become a reference method to estimate growth in mycoplasmas that acidify the medium when it grows

like *M. pneumoniae*. As described by Yus et al (2009), the analysis of the whole absorbance spectrum of the pH indicator used in the medium (phenol red) showed two wavelengths that respond differently to pH, around 430 nm (yellow) and 560 nm (red). The ratio between these two absorbances correlates well with changes in pH produced by the active lactate metabolism of *M. pneumoniae* when it grows (medium changes from red to yellow). Based on this principle, this method is an indirect but straightforward method to measure growth. The original reference of this method is now indicated in page 6 line 137. Also, to clarify the principle of the method the sentence in page 6 137-142 has been modified as follows:

“This method estimates growth by measuring the change of absorbance (growth index= ratio 430 and 560 nm) in the culture medium, and relies on the fact that M. pneumoniae acidifies this medium when it is metabolically active. A decrease of pH in the medium, detected by an increase in the growth index, is therefore an indirect but straightforward method to measure growth (9).”

(3) Page 6, line 141-142: “we tested decreased concentrations of HEPES, detecting a measurable change in pH with 50mM HEPES (Fig. S1).” The data did not show the changes in pH indicators?

We totally agree with this reviewer that Fig. S1 does not directly show changes in pH. However, as stated in the previous response, changes in the growth index (that is represented in Fig. S1) correlates with changes in pH. To clarify this, we have modified the sentence as follows (now page 6, line 146):

“...we tested decreased concentrations of HEPES, detecting a measurable change in the growth index with 50mM HEPES (Fig. S1)”

(4) Page 6, line 141-144: The pH indicators in different mediums should change during the culture. You should judge the effect of different mediums by the degree of pH change, not simply by whether the pH index has changed.

As commented in the previous responses, the pH indicator actually changes during the culture (from red to yellow) due to the acidification of the medium. Measuring the ratio absorbance of 430 and 560 nm (growth index) is a quantitative method to measure the degree of these pH changes. To illustrate this, below we show

data described in Yus et al., 2009 of a representative experiment of *M. pneumoniae* cultures in 96-well plates (A), and the correlation between pH and growth index (B).

A) Typical *M. pneumoniae* growth experiment done in a 96-well format in a Tecan plate reader, using a pH indicator, phenol red, to monitor growth (changes from red to yellow as the medium acidifies because of fermentation).

B) The ratio of the pH indicator (Phenol red) absorbance at 430 versus 560 nm (“growth index”) correlates well with the pH, at the pH range of the experiments performed (5.7-7.7). Equation for calculation of pH from the growth index, estimated for the pH range for which there is linear relationship (pH 5.7-7.7, $R^2=0.9931$).

(5) Page 7, line 151: “.....and the analysis of the metabolic map of *M.pneumoniae*.....”
Where is this data?

The analysis and refinement of the metabolic network of *M. pneumoniae* has been previously published. The data is available in references (9) and (26) as indicated in page 7, line 152 (now line 156).

(6) Page 12, line 323-325: “these results show a minor impact in gene expression.....”
Whether this change will also affect the function of subsequent vaccines

We agree with this reviewer that we cannot rule out the possibility that these changes may have a negative impact for the production of some vaccines or other therapeutic products produced by *M. pneumoniae*. However, the changes observed are minor, suggesting at least the “potential” of this medium for production. In fact, experiments with this medium in the framework of the MycoSynVac project (<https://www.mycosynvac.eu/>), which aim to express specific recombinant proteins on the bacterial surface as a live vaccine, did not show a negative impact of expression (data not published). However, we agree that in other payloads it may have

an impact. To note this, we have modified this sentence as follows in page 12, line 326-329:

*“Although quality assessment of the final bacterial product is required, these results show a minor impact in gene expression, supporting vB13 medium as a suitable and potential formulation for the production of *M. pneumoniae*-based therapies.”*

(7) In this study, a serum-free medium was developed to replace the traditional medium. In the optimization process, whether there are still some natural ingredients in serum cannot be determined by artificial optimization?

The specific components in serum have not been fully identified, thus it is difficult to validate the specific contributions of natural ingredients of the serum. In addition, the concentration and combinatorial effects of these compounds are difficult to predict and assess by artificial optimization. The fact that our medium does not perform 100% as rich medium, suggests that there is still room for improvement, and we presume that transport proteins (such as lipoproteins) present in the serum are probably required for an efficient supply of lipids. In fact, lipid metabolism is by far the more limited biosynthetic pathway of *M. pneumoniae*, which entirely depends on external lipid sources. However, supplementation of these types of molecules would probably result in a medium containing components of animal origin, and increase considerably the cost of the medium in the case that these compounds would be commercially available. Therefore, although we agree that the identification of other natural ingredients needed from the serum is important, we believe that is difficult to achieve based on the metabolic network of *M. pneumoniae*, and probably incompatible with a medium suitable for industrial production.

April 3, 2023

Dr. Raul Burgos
Centre for Genomic Regulation (CRG). The Barcelona Institute of Science and Technology
Dr Aiguader 88
Barcelona, Barcelona 08003
Spain

Re: Spectrum04859-22R1 (Development of a serum-free medium to aid large-scale production of Mycoplasma-based therapies)

Dear Dr. Raul Burgos:

Your manuscript has been accepted, and I am forwarding it to the ASM Journals Department for publication. You will be notified when your proofs are ready to be viewed.

Sincerely,

Jing Han
Editor, Microbiology Spectrum
